# Reported Changes in Adolescent Psychosocial Functioning during the COVID-19 Outbreak

**Sabina Kapetanovic** [1,2,*], **Sevtap Gurdal** [1], **Birgitta Ander** [2] **and Emma Sorbring** [1]

[1]   Department of Social and Behavioral Studies, University West, 46132 Trollhättan, Sweden;
      sevtap.gurdal@hv.se (S.G.); emma.sorbring@hv.se (E.S.)
[2]   School of Health and Welfare, Jönköping University, 55318 Jönköping, Sweden; birgitta.ander@ju.se
[*]   Correspondence: sabina.kapetanovic@hv.se; Tel.: +46-704-851-666

**Abstract:** What effect the outbreak of the COVID-19 pandemic has had on adolescents' psychosocial functioning is currently unknown. Using the data of 1767 (50.2% female and 49.8 male) adolescents in Sweden, we discuss adolescents' thoughts and behaviors around the COVID-19 outbreak, as well as reported changes in substance use, everyday life, relations, victimization, and mental health during the outbreak. Results showed that (a) the majority of adolescents have been complying with regulations from the government; (b) although most adolescents did not report changes in their psychosocial functioning, a critical number reported more substance use, conflict with parents, less time spent with peers, and poorer control over their everyday life; and (c) the majority of adolescents have experienced less victimization, yet poorer mental health, during the COVID-19 outbreak. Adolescent girls and adolescents in distance schooling were likely to report negative changes in their psychosocial functioning during the COVID-19 outbreak. Based on these findings, we suggest that society should pay close attention to changes in adolescents' psychosocial functioning during times of crisis.

**Keywords:** crisis; adolescents; psychosocial functioning; mental health; distance education

---

## 1. Reported Changes in Adolescents' Psychosocial Functioning during the COVID-19 Outbreak

The COVID-19 outbreak has paralyzed the entire world. For many, not least for adolescents, drastic public health measures such as social distancing and the closure of educational institutions have been some of the most difficult challenges posed by the COVID-19 outbreak. Given that adolescents seem to be particularly vulnerable to mental health problems [1], it is possible that drastic changes to adolescents' everyday lives during COVID-19 may have adverse effects on adolescents' psychosocial functioning.

*1.1. Adolescents' Psychosocial Functioning in Crises*

Although many adolescents manage to navigate the transition to adulthood well, adolescence is usually considered a period of heightened risk for psychosocial problems and psychiatric disorders [1,2]. Experiencing times of crisis during such a critical developmental period of life can put a strain on adolescents' psychosocial functioning [3], which could, in turn, be harmful for their future development. Indeed, different precautions in light of the pandemic seem to affect adolescents in varied ways. Although these precautions are deemed necessary, scholars worry that distance schooling [4], along with social distancing [5], could have adverse impacts on adolescents' psychosocial functioning [3]. Indeed, adolescents, particularly high-school seniors, have reported experiencing distress and concern about the future due to being away from school and peers [6]. Other studies have shown that school closures can be linked to decreased physical activity, sleep problems, a less healthy diet, and increased screen time [7,8], which can sometimes lead to mental health problems, including depression [9]. Scholars also

worry that the pandemic may also lead to increased engagement in risky behaviors, such as substance use, as well as increased offline and online victimization among adolescents [3]. Moreover, a study from China recently showed high levels (18.9%) of anxiety and depression among students in the Hubei province after an approximate lockdown of 30 days [10]. By contrast, a Swiss longitudinal study, which started two years before the pandemic began, showed that there was an increase in adolescents' anger and stress levels, but a decrease in internalizing symptoms such as depression, during the pandemic [11]. The same study revealed, however, that those who had experienced pre-pandemic social stressors (e.g., bullying victimization, stressful life events, and feelings of social exclusion) were more likely to experience more emotional distress during the pandemic. Girls, more often than boys, reported emotional distress and hopelessness. Given that adolescent girls, in general, report higher levels of mental health problems [12], experiencing times of crises could be particularly harmful for adolescent girls' psychosocial functioning.

Adolescence is considered one of the most critical developmental periods in human life [13]. During this period in life, adolescents strive for autonomy from parents and seek to spend more time with friends [14], which is why socializing with peers and engaging in social activities is critical to their psychosocial development. A sudden disruption to their routines, during which adolescents are forced to spend more time with family and less time with peers, could have both positive and negative outcomes in terms of adolescents' psychosocial functioning. For example, staying at home could create more opportunities to do fun and enjoyable things with family, which may result in better parent–child relationships [15]. In addition, parents may have a better opportunity to monitor their children, which could decrease the risk of children engaging in harmful activities such as substance use [16], which generally increases during adolescence [17,18]. On the other hand, an increased amount of time spent at home with family can also lead to more conflict between youth and their parents and to harsher discipline measures and control by parents [19], which could have an adverse effect on adolescents' psychosocial functioning [20].

## 1.2. Cultural Context and the Current Study

Due to the COVID-19 pandemic, countries have taken precautions to control the transmission of coronavirus. The Swedish strategy has included a great focus on the individual's own responsibility to contain the virus. The government has imposed recommendations to maintain a physical distance (also called social distancing), to wash hands regularly, to stay at home when experiencing any symptoms at all, to avoid physical contact with people older than 70, to avoid unnecessary traveling, and, when possible, to work from home [21,22]. Contrary to many other countries, no lockdown or obligatory face mask use has been implemented in the Swedish strategy. Moreover, although with measures implemented to avoid crowds of people, restaurants, pubs, local transportation services, and shopping malls have mostly remained open during the pandemic [22]. One explanation behind the Swedish approach is that Swedish residents have a high level of trust in government agencies and, therefore, generally do as they are told [23]. The main effort in Sweden has been to limit the spread among people, to have the right medical resources to help those infected, to limit the impact on critical services such as healthcare and the police, and to ease the impact on people, but also on businesses, with crisis packages available to help organizations and to ensure that people's jobs are not at risk [21]. While many countries around the world have implemented strict domestic lockdown policies, including distance schooling for children of all ages, in Sweden distance schooling has been implemented primarily in upper secondary education and universities, which moved to distance education on the 17 March 2020 [21]. However, students in lower secondary schools have had regular schooling throughout the pandemic so far and not turned to distance education at all.

At almost the same time, on 29 March, the restriction of having no more than 50 people at any event (both public and official) was introduced in Sweden. For adolescents, this was a particularly noticeable change due to the restrictions imposed on graduation ceremonies, leisure-time activities, concerts, festivals, clubs, etc. Professionals have raised concerns about the effect of the pandemic on

adolescents' psychosocial functioning, particularly due to the precautions of distance schooling and social distancing. In this paper, we aimed to show adolescents' thoughts and behaviors around the COVID-19 outbreak as well as to report changes in substance use, everyday life, relations, victimization, and mental health during the pandemic. These results are important given that adolescence is a time of important biopsychosocial changes [13], making adolescents a vulnerable group during times of crisis [3]. We hypothesized (a) that adolescents would report an increase in substance use and victimization [3], more conflicts or quality time spent with parents [19], less time spent with peers [6], and poorer everyday life and mental health [9]; (b) that particularly girls [11,12] would be subject to poorer psychosocial functioning during COVID-19; and (c) although some studies have shown mixed results [11], that adolescents who are engaged in distance schooling [7,8] would report poorer psychosocial functioning during COVID-19. Given the paucity of research on adolescent perceptions of the restrictions during COVID-19, no specific hypotheses regarding adolescents' thoughts and behaviors around the COVID-19 outbreak were made.

## 2. Material and Methods

### 2.1. Procedure

The project COVIDung was designed to recruit adolescents of ages 15 to 19 to study adolescents' social relations, individual characteristics, and psychosocial changes during the COVID-19 outbreak. Participant recruitment and survey completion through an online survey occurred between 8 June and 7 July 2020. The dates corresponded with the end of the semester and the beginning of summer vacation for students both at lower and upper secondary schools. Advertisements for the study were posted mainly on Instagram (70%), as well as Facebook and Twitter. Before answering the survey, participants were informed about the study and their rights as participants. Individuals who were interested in participating followed the link to the survey. Survey statistics revealed that we reached almost 266,000 individuals (out of about 400,000 possible in the targeted age-group) with the advertisements and that 7217 individuals clicked on the link to the survey. We used a secure online platform for the survey, administrated by a company specializing in research surveys. The survey took approximately 15–20 min to finish, and participants could use either a smartphone or a computer. All procedures performed in the study were in accordance with the ethical standards of the institutional research committee (ethical committee at Jönköping University (number 20.6)) and with the 1964 Helsinki declaration and its later amendments, or comparable ethical standards. Informed consent was obtained from all individual participants included in the study. The datasets generated and/or analyzed during the current study are available from the corresponding author on reasonable request.

### 2.2. Participants

In total, 1818 adolescents completed the survey. Out of that total, 51 adolescents were excluded from the sample because they reported being younger than 15 or older than 19 years of age. Thus, the final sample consisted of 1767 adolescents (50.2% females and 49.8% males), born 2001–2005 (*Md* = 2003), and living in Sweden. About 76% were students in upper secondary school (ages 17–19), and 24% were students in lower secondary school (ages 15–16). In our sample, 99% of students at the upper secondary school level were engaged in distance education, while only 14% of students at the lower secondary school had it (probably due to personal needs). In total this means that about 80% of youth in our sample moved to distance education after mid-March. Furthermore, the participants came from different Swedish counties, although the majority (56%) came from counties where larger cities are located. Most of the participants lived with both their parents (77%) and had parents who held a full-time job (mothers 80%; fathers 86%).

## 2.3. Measures

Adolescents' Thoughts and Behaviors around the COVID-19 Outbreak. The instrument was developed by [24] assessing confidence in the government (2 items), hope for the future (1 item), and compliance with rules (2 items), all in relation to the COVID-19 outbreak, measured on a 4-point scale ranging from 1 (do not agree at all) to 4 (agree completely). Internal consistency of the scale was $\alpha = 0.55$.

Changes in Adolescents' Substance Use, Relations, and Everyday Lives. Adolescents reported changes concerning substance use, such as alcohol and tobacco (4 items); relations with family and friends (4 items) and everyday life situations (3 items). All items were measured on a 5-point Likert scale ranging from 1 (decreased a lot) to 5 (increased a lot). The participants could also respond 0 for "I did not do this before the outbreak and have not started".

Changes in Adolescents' Victimization. Changes in victimization were assessed with five items from the Swedish Crime Survey [25]. The items covered physical violence, threats, and sexual harassment (3 items) and online victimization (2 items), measured on a 5-point scale ranging from 1 (decreased a lot) to 5 (increased a lot). Internal consistency of the scale was $\alpha = 0.92$.

Changes in Adolescents' Mental Health. Nine items from the Experiences Related to COVID-19 instrument [24] were used to assess adolescents' reported changes in sleep, stress, satisfaction, loneliness, anger, depression, and anxiety. The items were measured on a 4-point scale ranging from 1 (don't agree at all) to 4 (agree completely). Internal consistency of the scale was $\alpha = 0.82$.

Demographics. We used adolescents' gender and schooling situation as demographic variables. Adolescents' gender was coded 0 = boys and 1 = girls, and schooling was coded 0 = regular schooling and 1 = distance schooling.

## 2.4. Statistical Analyses

We used SPSS 25 to conduct statistical analyses. Descriptive analyses were conducted to show the frequencies of adolescents' thoughts and behaviors around the COVID-19 situation and reported changes in psychosocial functioning. We dichotomized all ordinal scale variables. The alternatives "somewhat disagree" and "strongly disagree" were categorized as "Disagree" and the alternatives "somewhat agree" and "strongly agree" were categorized as "Agree". The alternatives "decreased a little" and "decreased a lot" were categorized as "Decreased", and the alternatives "increased a little" and "increased a lot" were categorized as "Increased". Moreover, we conducted Pearson's chi-square tests to compare boys and girls, as well as adolescents with regular schooling and distance schooling, in terms of their thoughts and behaviors around the COVID-19 situation and reported changes in psychosocial functioning.

## 3. Results

### 3.1. Adolescents' Thoughts and Behaviors around the COVID-19 Outbreak

We asked the adolescents about their thoughts and behaviors around the COVID-19 outbreak. As shown in Figure 1, the majority of the adolescents (65.9–92.2%) had positive views of how the government is handling the situation, and a good outlook toward the future and the virus resolving over time. Most of the adolescents (77.8–92.3%) also reported that they complied with the rules and suggestions of the government and health care system to contain the virus and found it easy to do so.

Comparison of boys and girls revealed that girls were more likely than boys to report having confidence in the government handling the COVID-19 response (72.1% vs. 59.5%, $p < 0.001$), and complying with rules and suggestions of the government and health care system to contain the virus (91.2% vs. 82.4%, $p < 0.001$). Boys were, however, more likely than girls to report being hopeful that the virus will resolve over time (86.3% vs. 80.2%, $p < 0.001$). Moreover, adolescents with distance education were more likely than adolescents with regular education to report complying with rules and regulations (88.1% vs. 82% $p < 0.001$).

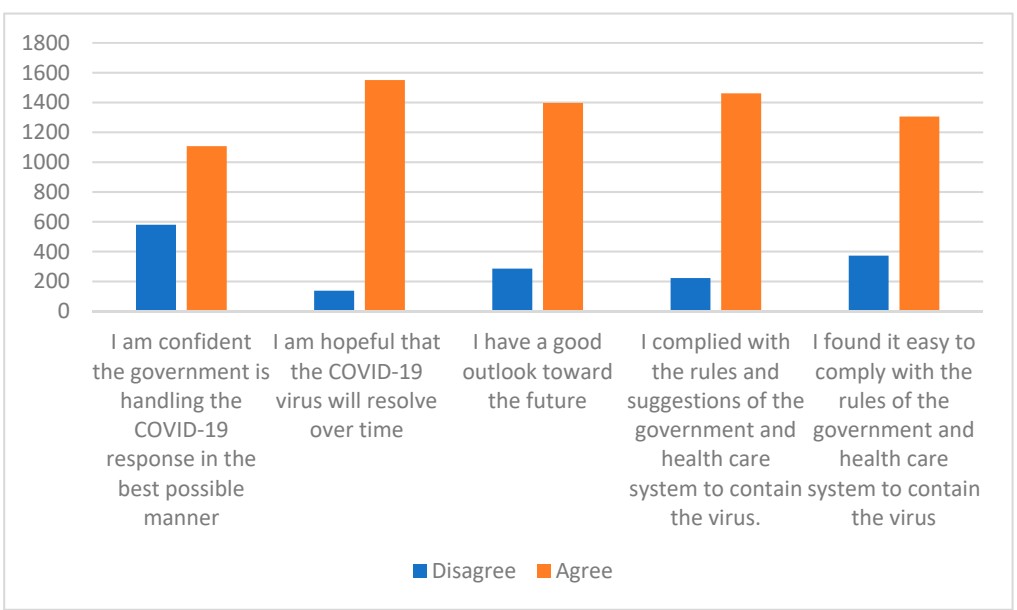

**Figure 1.** Adolescents' thoughts and behaviors around the COVID-19 outbreak.

### 3.2. Reported Changes in Adolescents' Substance Use, Relations, and Everyday Lives

As shown in Table 1, most of the adolescents reported that their substance use, relations with family and friends, and everyday lives were relatively unchanged in comparison to the period before the COVID-19 outbreak. However, although more than 10% of the adolescents stated that their alcohol use and inebriation had decreased, 17.6% and 14.1% of adolescents, respectively, reported that their alcohol use and inebriation had increased. A total of 30.1% of adolescents reported a decrease in spending time with family doing fun things, while 29.9% of adolescents indicated that conflict with parents had increased. In addition, 13.6% of adolescents reported that their hanging out on the streets without parents' knowledge had decreased, and a total of 49.6% of adolescents reported a decrease in meeting with friends offline. Finally, a total of 47.7% of adolescents indicated that they had more time for things they had not had time for before. However, 39.8 and 35.9%, respectively, reported a decrease in being in sync with tasks at school and having control of their everyday life.

Comparison of adolescents who reported changes in their substance use ($n = 95–494$), relations ($n = 619–1126$), and everyday life ($n = 97–1024$) showed that boys were more likely than girls to report a decrease in their use of narcotics (54.0% vs. 32.4%, $p < 0.05$) and in meeting with friends offline (77.5% vs. 66.1%, $p < 0.001$). Girls, however, were more likely than boys to report an increase in having conflicts with parents (84.8% vs. 74.2%, $p < 0.05$).

Furthermore, adolescents who had distance education were more likely than adolescents who had regular education to report decreases in alcohol use (43.2% vs. 28.4%, $p < 0.05$), meeting with friends offline (75.2% vs. 55.0%, $p < 0.001$), and being in sync with tasks at school (64.9% vs. 54.7%, $p < 0.05$). Adolescents who had distance education were more likely than adolescents who had regular education to report having conflicts with parents (81.1% vs. 72.7%, $p < 0.05$), while adolescents who had regular education were more likely than adolescents who had distance education to report a decrease in time spent with family doing fun things (60.4% vs. 51.5%, $p < 0.05$) and having less control of their everyday life situation (69.6% vs. 56.0%, $p < 0.05$).

**Table 1.** Reported changes in adolescents' substance use, relations with family and friends, and everyday life situation.

| Item | | Frequency | | | |
|---|---|---|---|---|---|
| | *n* | **Have Not Done It Before** *n* **(%)** | **Decreased** *n* **(%)** | **Unchanged** *n* **(%)** | **Increased** *n* **(%)** |
| **Substance use** | | | | | |
| Smoking cigarettes | 1642 | 1325 (80.7) | 97 (5.9) | 112 (6.8) | 108 (6.6) |
| Drinking alcohol | 1644 | 765 (46.5) | 204 (12.4) | 385 (23.4) | 290 (17.6) |
| Getting inebriated | 1643 | 921 (5.1) | 202 (12.3) | 288 (17.5) | 232 (14.1) |
| Using narcotics | 1640 | 1485 (90.5) | 45 (2.7) | 58 (3.5) | 52 (3.2) |
| **Relations with family and friends** | | | | | |
| Spending time with family doing fun things | 1644 | 65 (4.0) | 495 (30.1) | 651 (39.6) | 433 (26.3) |
| Having conflicts with my parents | 1646 | 337 (20.5) | 127 (7.7) | 690 (41.9) | 492 (29.9) |
| Hanging out on the streets without my parents' knowledge | 1642 | 620 (37.8) | 223 (13.6) | 622 (37.9) | 177 (10.8) |
| Meeting with friends offline | 1642 | 60 (3.7) | 815 (49.6) | 456 (27.8) | 311 (18.9) |
| **Everyday life situation** | | | | | |
| Having time for things that I have not had before | 1642 | 46 (2.8) | 192 (11.7) | 620 (37.8) | 784 (47.7) |
| In phase with tasks at school | 1641 | 26 (1.6) | 653 (39.8) | 591 (36.0) | 371 (22.6) |
| Having control of my everyday life | 1640 | 17 (1.0) | 589 (35.9) | 600 (36.6) | 434 (26.5) |

## 3.3. Reported Changes in Adolescents' Victimization

We asked the adolescents whether they experienced decreases or increases in the level of offline and online victimization. As shown in Figure 2, most adolescents reported being neither more nor less victimized after the COVID-19 outbreak than before. In terms of any kind of changes, then, the sample tended to report more decreases than increases in offline and online victimization.

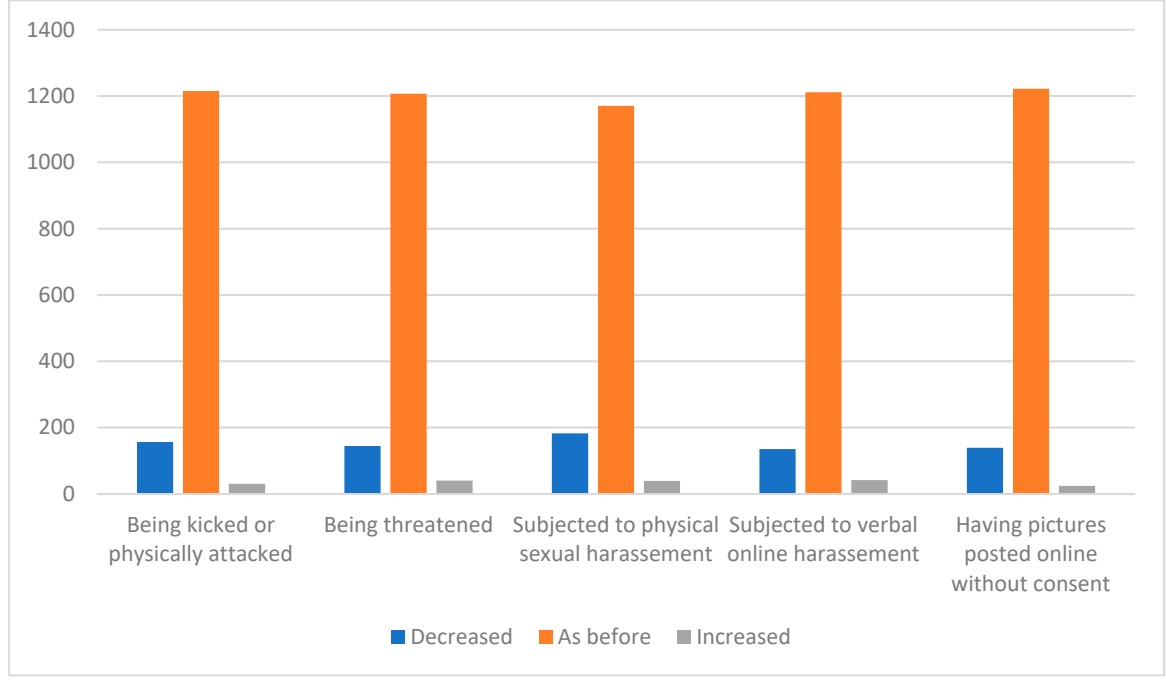

**Figure 2.** Reported changes in adolescents' offline and online victimization.

Out of those who reported changes in victimization (*n* = 163–221), adolescents who had distance education were more likely than adolescents who had regular education to report a decrease in being

kicked or physically attacked (85.5% vs. 71.4%, *p* < 0.05) and being threatened (83.5% vs. 63.8%, *p* < 0.05). No significant difference between genders in victimization was found.

### 3.4. Reported Changes in Adolescents' Mental Health

As shown in Table 2, the majority of the adolescents reported that they had more internalizing symptoms, such as being sad, anxious, and lonely, and externalizing symptoms, such as being angry and arguing, now compared to before the COVID19 outbreak.

**Table 2.** Reported changes in adolescents' mental health.

| Item | N | Strongly Agree *n* (%) | Agree *n* (%) | Disagree *n* (%) | Strongly Disagree *n* (%) |
|---|---|---|---|---|---|
| I sleep about as well now as I did before the COVID-19 outbreak | 1651 | 440 (26.7) | 615 (27.2) | 398 (24.1) | 198 (12.0) |
| I sleep more now at unusual times than I did before the outbreak | 1647 | 367 (22.3) | 633 (38.4) | 379 (23.0) | 268 (15.3) |
| I feel more anxious now than I did before the outbreak | 1646 | 213 (12.9) | 632 (38.4) | 509 (30.9) | 292 (17.7) |
| I feel more sad/depressed now than I did before the outbreak | 1648 | 301 (18.3) | 535 (32.5) | 485 (29.4) | 327 (19.8) |
| I feel more angry now than I did before the outbreak | 1647 | 181 (11.0) | 488 (29.6) | 596 (36.2) | 382 (23.2) |
| I get in more arguments now than I did before the outbreak | 1646 | 103 (6.3) | 370 (22.5) | 655 (39.8) | 518 (31.5) |
| I feel more lonely now than I did before the outbreak | 1648 | 403 (24.5) | 578 (35.1) | 357 (22.8) | 292 (17.7) |
| I feel less stressed now than before the outbreak | 1645 | 140 (8.5) | 409 (24.9) | 652 (39.6) | 444 (27.0) |
| I feel more satisfied now than before the outbreak | 1642 | 75 (4.6) | 341 (20.8) | 813 (49.5) | 413 (25.2) |

Comparison of boys and girls showed that girls, overall, had poorer mental health during the COVID-19 outbreak. Males, to a greater extent than girls, reported sleeping about as well now as before the outbreak (69.4% vs. 58.9%, *p* < 0.001). Girls, to a higher degree than males, reported having poorer sleep (41.1% vs. 30.6%, *p* < 0.001), being more anxious (63.3% vs. 38.2%, *p* < 0.001), sad/depressed (59.7% vs. 41.1%, *p* < 0.001), angry (48.4% vs. 32.2%, *p* < 0.001), getting in more arguments (32.3% vs. 24.6%, *p* < 0.001), feeling more lonely (66.7% vs. 52.0%, *p* < 0.001), stressed (701% vs. 63.0%, *p* < 0.001), and less satisfied (79.3% vs. 69.7%, *p* < 0.001) now than before the outbreak.

Some differences were also found when adolescents who had regular education were compared with adolescents who had distance education. Adolescents who had regular education in comparison to those who had distance education reported being less satisfied with their situation (79.9% vs. 73.3%, *p* < 0.05). However, adolescents that had distance education, to a greater extent than those who had regular education, reported poorer sleep (38.6% vs. 25.7%, *p* < 0.001), sleeping at more unusual times (65.4% vas 42.7%, *p* < 0.001), being more anxious (52.9% vs. 44.4%, *p* < 0.05), feeling more sad/depressed (53.6% vs. 38.7%, *p* < 0.001), getting in more arguments (29.9% vs. 22.9%, *p* < 0.05), and feeling more lonely (64.0% vs. 41.5%, *p* < 0.001).

## 4. Discussion

The aim of this study was to capture adolescents' thoughts and behaviors in relation to the COVID-19 outbreak, as well as reported changes in substance use, everyday life, relations, victimization, and mental health during the outbreak.

The overall results in our study indicate that adolescents complied with regulations and had confidence in the government's handling of the COVID-19 situation. There were, however, reported effects on their everyday lives and relationships. For example, about 50% of the adolescents reported spending less time with friends and more time with family after the outbreak than before. One-third of adolescents reported a decrease in spending fun time with the family, and another third reported an increase in conflicts with parents. Although most of the adolescents did not report any changes in their substance use and victimization, for some adolescents the restrictions due to COVID-19 have resulted in more alcohol use, whereas for others it led to less alcohol use. Notwithstanding that many adolescents reported having more opportunities to do things they had not had time for before the outbreak, adolescents reported having less control over their everyday life, having trouble being in sync with tasks at school, and exhibited worrying tendencies of mental distress, including anxiety, loneliness, and stress. Negative changes during the COVID-19 pandemic particularly seemed to apply to girls and adolescents who have had distance education.

Being in transition to adulthood is not always easy. While some adolescents go through adolescence without hardship, for some adolescents the period of adolescence can be a time of challenge and distress [2,13]. Adolescents are therefore vulnerable and particularly could be exposed to potentially harmful consequences when their everyday life and environment suddenly changes [3]. Even though adolescence is one of the most critical periods in life [1], little attention has been given to adolescents and their psychosocial functioning during the COVID-19 pandemic. Indeed, scholars have raised important questions about the well-being and mental health of adolescents during the crisis [3,5,26]; however, little attention has been paid to adolescents' situation, both in media and by governmental policies. This is important considering that, although adolescents are not the primary risk group for spreading of the COVID-19 disease, they have been particularly restricted in their everyday life given that their social environment abruptly changed. This has all been for the benefit of adults. They are to stay at home, avoid contact with peers, and find ways to cope with school and daily life. They are also compliant. According to our results, this sample of adolescents reported having confidence in the government's handling of the situation and that they comply with the rules and regulations, including spending less time with peers and more at home. But at what cost?

As shown in our study, for many adolescents, spending more time at home included having more conflicts with parents and less time for fun things with family. These relational changes are important as they could impact adolescents' development, not the least in how they cope with difficulties during times of crisis [4,26]. For example, most of the adolescents in our study reported poorer mental health now than before the outbreak. This particularly seemed to be the case for girls and for adolescents who have had distance schooling during COVID-19 crisis. These results corroborate the findings from other studies showing that female gender [11] and distance schooling [6] are important risk factors during the time of crisis, such as a pandemic. Indeed, adolescents' school context was suddenly reduced to distance education at home and less social support from teachers and peers. Social support plays a critical role for development of resilience [27]. When such a critical element of adolescent lives is lacking, as it does when schools are closed, it may have negative impacts on adolescent psychosocial adjustment [28,29]. This may be especially true during times of crisis, such as a pandemic [30]. In addition, lacking the structure that schools provide can result in sleep problems, a less healthy diet, and more screen time [7,8], which in turn may have negative impacts on adolescent mental health [9]. As scholars suggest [3], adolescents are likely to engage in substance use as a means of coping with difficult situations. Although most of the adolescents in our study did not report changes in their substance use, some did report increases in their substance use, while others decreased their use of alcohol and drugs. As strong parent–child relationships and parental monitoring are some

of the central protective factors for substance use engagement [16,31], it could be speculated that adolescents who perceive having poorer relations with parents during the time of the pandemic might also increase their substance use, potentially as means of coping with the current situation. Moreover, given that peers provide strong support in terms of well-being [32], another potential explanation is that adolescents may perceive more mental health problems related to reduced socializing with peers. Such hypotheses should be investigated in future research.

One positive aspect of changes in adolescents' everyday lives is that the reports of peer victimization were lower rather than higher during this period of the pandemic. As seen in the results, particularly adolescents who had distance education reported less physical and sexual victimization during pandemics. One explanation of such a trend could be that context plays a role. As victimization among adolescents often takes place in school or peer contexts [33], and continues online [34], being outside of such contexts could, in a sense, buffer the risk of being subjected to victimization. However, there are also groups of adolescents who experienced increases in victimization. Corroborating other scholars' concern [3], we worry that social distancing and lack of social support that schools and their classmates often can provide [35] could be harmful for adolescents' future development. More in-depth research needs to be done to understand the risk and protective factors in terms of adolescents' psychosocial functioning during periods of crisis.

There are some limitations in this study. Because of the cross-sectional nature of the data, we cannot control for the previous levels of adolescents' psychosocial functioning or make inferences of causality. Next, the participants were recruited through the internet and, more specifically, social media. Such a design carries a risk for selection bias because it limits the population to internet users. However, 98% of the population in Sweden has access to internet at home, and the majority of adolescents are frequent social media users [36], which makes the risk of selection bias small. Furthermore, we did not investigate the possible correlates of adolescent reported psychosocial changes. It is, nevertheless, of great importance to understand the possible risk and protective factors pertaining to reported changes in adolescent psychosocial functioning. Therefore, in future studies with the same sample, we will focus on parent–child relationships, peer relations, and adolescent personality traits as possible correlates of reported changes in their psychosocial functioning. Some strengths of the current study include having a large and representative sample and a focus on adolescents, who are in the crucial transitional stage of development [13] now, during the period of crisis.

## 5. Conclusions

In conclusion, the descriptive results in this study indicate that adolescents experienced negative relational changes and poorer mental health during the COVID-19 crisis. They report being compliant with the rules and regulations at the cost of their psychosocial functioning. The message we want to send is that society has to take action in times of crises and social distancing in order to make sure that adolescents have all the support they need to be able to handle the challenges they face. Adolescents need to be engaged in creating governmental policies and have opportunities to be guided into the future. When everyday school support is harder to reach for adolescents, schools need to get resources not only to provide education from a distance, but also to maintain and expand school counselling and mental health support services. In case of future periods of social distancing and distance education, government policies need to act fast and with great concern about adolescents' well-being, now and in the future.

**Author Contributions:** Conceptualization, methodology, original draft preparation and writing S.K., S.G., B.A., E.S.; data analyses, S.K., data interpretation, all authors. All authors have read and agreed to the published version of the manuscript.

**Funding:** This research received no external funding.

**Conflicts of Interest:** The authors declare no conflict of interest.

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
