# Peer review of "Reported Changes in Adolescent Psychosocial Functioning during the COVID-19 Outbreak"

_adolescents, doi:10.3390/adolescents1010002_

Round 1

Reviewer 1 Report

Attractive title.

Correct summary that meets your objective. The first function of the abstract is to explain the reason for the chosen title. This will allow the reader to decide if the research topic is interesting enough to continue reading or not. A well-written summary can help convince readers that your work is interesting and important and keep them reading it.

This summary fulfills this function.

Beware of APA standards, check that a comma is missing after et al., In all citations.

 Regarding the review of the state of the question, it is totally pertinent and adequate.

An important aspect that the article highlights is the governmental responsibility in paying attention to the adolescent population. It has been believed that young people adapt well to any change and that it has no consequences and it has been shown with this article that there have been aspects that have improved, such as family relationships in adolescents who did not usually have parental conflicts but nevertheless those who had conflicts previously, confinement has aggravated them.

The number of the sample is quite representative and the bias that could be due to filling in the questionnaires through social networks, I think it has been minimal, because currently, most young people handle everything with social networks.

The statistics for the object of the study are adequate and the conclusions are revealing of how we have not dealt with adolescents because we believe that it was a population without risk when it has been verified that social issues with peers are fundamental at this stage.

An interesting article.

Author Response

Thank you so much for your review. 

Reviewer 2 Report

The focus on this paper is the negative impact of protective measures in front of the Covid-19 Outbreak, on psychosocial functioning of adolescents.  1789 Swiss secondary students were surveyed, and data were collected on a) adolescents’ thoughts and behavior about the outbreak, b) changes in adolescents’ substance use, relations, and every day lifes, c) changes in adolescents’ victimization, and d) changes in adolescents’ mental health.

The research question is relevant and interesting, although its presentation is somewhat confusing. Arguments should be better organized along the text, and coherence between the antecedents synthetized in the Introduction, Material and methods section, and the interpretation of data obtained should strengthen. More precisely:

  • The first paragraph in the Introduction contains some ideas that seem out of place:
    • Presumed adverse effects of social distancing are unnecessarily advanced, a theme that is more deeply exposed in section 1. Adolescents’ psychosocial functioning in crises.
    • The aim of the study, which is introduced at the end of the paragraph, should be integrated at the end of the Introduction once the antecedents of the research topic have been exposed.
  • The first paragraph in the section 1. Adolescents’ psychosocial functioning in crises would be better placed at the in end of the section since the ideas exposed in that paragraph (regarding consequences of the Covid-19 Outbreak for family relations) are more concrete that those exposed in the second paragraph in the same section. This second paragraph begins mentioning transitional characteristics of adolescence and follows with additional disturbances associated with the Covid-19 Outbreak. Also, in this paragraph, the term “demonstrate” seems excessive for a descriptive study.
  • The connection between the data described on the incidence of increased diseases and risk factors in general population with consequences for adolescents is not clearly established.
  • The hypotheses posed for consequences of social distancing are not coherent with antecedents exposed in the Introduction where “both positive and negative outcomes in terms of adolescents’ psychosocial functioning” are considered. No hypotheses were established concerning adolescents’ thoughts and behaviors, and changes in substance use, peer relations, victimization and everyday lifes of adolescents.
  • The number of participants indicated in the Abstract does not match with that informed in the 2. Participants subsection.
  • With respect to Results section, a lot of information in the text overlaps with that given in the tables.
  • The antecedents exposed at the beginning of the Discussion are redundant with respect to those previously included in the Introduction.
  • The concluding remark “Their concerns were valid” would be more appropriately situated at the subsection 1. Limitations and Conclusions.
  • Arguments regarding “Why are these results important?” would be more appropriately placed in the Introduction, where the relevance of the issue is to be found.
  • Some relevant findings of the study are not discussed: disparities in complying rules and positive/negative changes registered, increased/decreased alcohol use, increased conflicts with parents…
  • The expected impact of social distancing for adolescents’ development and mental health should be further explained in the Discussion. Some ideas on this regard were included in the text: “…losing access to arenas that previously offered knowledge development and social interaction…”, “…can put a strain on adolescents’ psychosocial functioning”, “decreased physical activity, sleep problems, a less healthy diet, and more screen time” …
  • The last two paragraphs regarding statement on ethical standards and subsequent availability of the data, would be better placed in the Procedure subsection.

Some other issues:

  • The synthesis of results in the Abstract is not comprehensive enough.
  • Reference List and Citations should be adapted to MDPI’s style.
  • Journal article references are incomplete.
  • Expressions such as “data of N = 1789”, “n = 51” …should be reviewed.
  • The following references are recommended regarding the role of social support in transitional life events:

Martínez-López, Z., Tinajero, C., Rodríguez, M. S., & Páramo, M. F. (2019). Perceived social support and university adjustment among Spanish college students. European Journal of Psychology and Educational Research, 2(1), 21-30. https://doi.org/10.12973/ejper.2.1.21

Rodríguez, M. S., Tinajero, C., & Páramo, M. F. (2017). Pre-entry characteristics, perceived social support, adjustment and academic achievement in first-year Spanish university students: A path model. The Journal of Psychology, 151(8), 722-738. https://doi.org/10.1080/00223980.2017.1372351

Author Response

Thank you so much for your review, comments and suggestions which, we believe, have helped us to improve the quality of the manuscript. Please see attached file with our comments.
